# Translesion Synthesis or Repair by Specialized DNA Polymerases Limits Excessive Genomic Instability upon Replication Stress

**DOI:** 10.3390/ijms22083924

**Published:** 2021-04-10

**Authors:** Domenico Maiorano, Jana El Etri, Camille Franchet, Jean-Sébastien Hoffmann

**Affiliations:** 1Institute of Human Genetics, UMR9002, CNRS-University of Montpellier, 34396 Montpellier, France; domenico.maiorano@igh.cnrs.fr (D.M.); jana.eletri@igh.cnrs.fr (J.E.E.); 2Laboratoire D’Excellence Toulouse Cancer (TOUCAN), Laboratoire de Pathologie, Institut Universitaire du Cancer-Toulouse, Oncopole, 1 Avenue Irène-Joliot-Curie, 31059 Toulouse, France; franchet.camille@iuct-oncopole.fr

**Keywords:** genome instability, replicative stress, specialized DNA polymerases, translesion synthesis (TLS), Rad18, DSB repair, TMEJ, Pol theta, POLQ

## Abstract

DNA can experience “replication stress”, an important source of genome instability, induced by various external or endogenous impediments that slow down or stall DNA synthesis. While genome instability is largely documented to favor both tumor formation and heterogeneity, as well as drug resistance, conversely, excessive instability appears to suppress tumorigenesis and is associated with improved prognosis. These findings support the view that karyotypic diversity, necessary to adapt to selective pressures, may be limited in tumors so as to reduce the risk of excessive instability. This review aims to highlight the contribution of specialized DNA polymerases in limiting extreme genetic instability by allowing DNA replication to occur even in the presence of DNA damage, to either avoid broken forks or favor their repair after collapse. These mechanisms and their key regulators Rad18 and Polθ not only offer diversity and evolutionary advantage by increasing mutagenic events, but also provide cancer cells with a way to escape anti-cancer therapies that target replication forks.

## 1. Introduction

Replication stress (RS) is frequently observed in cancer cells. It is manifested by the persistence of stalled and collapsed forks, as well as by under-replicated DNA, which can be a source of chromosomal breakage and consequent chromosome instability. RS has been proposed to generate a permanent sub-population of cellular variants that, upon selection, could act as a driving mechanism of tumor heterogeneity, development, and drug resistance [1]. This clonal evolution can result from multiple forms of selective pressure, allowing some mutant subclones to proliferate while others become extinct. While genomic instability is generally associated with poor prognosis, excessive karyotypic instability is deleterious for cell fitness and correlates with improved survival outcome, supporting the existence of an appropriate threshold of genome instability for limiting extremely risky RS [2].

Preservation of genome integrity and the required adaptation to genotoxic stresses are two key elements for ensuring both cell survival and evolution of multicellular organisms. Accurate replication of undamaged DNA involves the action of the replicative high-fidelity DNA polymerases Polδ and ε which have proofreading ability (Lujan et al., 2016). When these enzymes encounter DNA distortions or persistent base modifications produced by endogenous or environmental insults, they frequently stall because of their high degree of selectivity and inability to faithfully insert a base opposite most lesions. To avoid aberrant cell cycle arrest induced by replication fork blockage, stalled DNA polymerases need to be transiently assisted by specialized, translesion synthesis DNA polymerases (TLS Pols), capable of bypassing the lesion. This process is less stringent and more mutagenic compared to the accurate DNA polymerization by replicative Pols, but it allows cells to tolerate structural DNA perturbations, a necessary “flexibility” [3].

Other specialized DNA polymerases are involved in the repair of double-stranded DNA breaks (DSBs), which are highly toxic. DSBs are usually repaired by the non-homologous end joining (NHEJ) pathway, which joins the two ends of a DNA break with limited processing of the broken ends, or by homologous recombination (HR), a conservative repair pathway where the sister chromatid is used as a template for restoring genetic information. HR also acts in the reactivation of replication forks blocked during chromosome duplication, a frequent event that occurs even during normal cell proliferation. Impairment of either of these repair pathways—especially HR impairment due to deficiency in BRCA1 or BRCA2—inevitably requires backup DNA repair pathways to solve replication-associated DNA breaks or gaps. These rescue mechanisms, which maintain cell survival, are generally error-prone and therefore entail a mutagenic cost. One important backup repair actor is the A-family specialized DNA polymerase theta (Polθ, gene name POLQ), which executes the so-called theta-mediated end joining (TMEJ) [4,5]. This represents an alternative form of end-joining directly competing with HR by roughly “gluing” DNA breaks in a mutagenic manner, notably in BRCA-deficient cancer cells where genome alterations result from TMEJ [6]. It has become apparent that HR-deficient cells have a greater reliance on the activity and expression of POLQ. This represents the main “Achilles heels”, which could putatively be targeted to kill cancer cells by a principle known as synthetic lethality [7]

In this review, we will discuss the contribution of the specialized or alternative DNA polymerases, to limit RS and genome instability. These include a group of enzymes characterized by their infidelity in replicating undamaged DNA and their ability to contribute to DNA transactions, such as translesion synthesis (TLS) or TMEJ DNA repair synthesis in response to genotoxic stress which could balance the evolutionary advantages associated with genomic instability against the fitness costs caused by it. This provides insight into the importance of these pathways to avoid breakage or repair of broken forks after collapse, while allowing diversity and evolutionary advantage by increasing mutagenic events and contributing to some aspects of therapeutic resistance. We will particularly focus on the critical actors of these two processes, Rad18 and Polθ, which represent promising targets in cancer therapy.

## 2. The Crucial Role of TLS to Tolerate Structural Impediments during DNA Replication

### 2.1. TLS at Arrested Forks

DNA replication is not a linear and continuous process. Replication forks frequently encounter obstacles that slow down or even halt the progression of DNA synthesis, leading to RS. The obstacles can be of endogenous nature, generated by the cellular metabolism, in addition to those generated by exposure to environmental polluting chemicals and chemotherapeutic agents. Although cells have evolved efficient DNA repair mechanisms, DNA lesions may escape repair and interfere with DNA synthesis. This depends upon the amount of DNA damage as well as its nature. For example, UV-C light generates two types of DNA lesions, *cis–syn* cyclobutane pyrimidine dimers (CPDs) and (6-4) pyrimidine–pyrimidone photoproducts [(6-4)PPs]. However, CPDs are removed more slowly than [(6-4)PPs] by the nucleotide excision repair (NER) system [8], leading to their persistence in the S-phase. The consequences of RS depend upon the nature of the obstacle encountered by the DNA synthesis machinery, named the replisome, recently reviewed in [9]. One such consequence is the uncoupling of replicative polymerase and helicase movements. This process leads to the arrest of replicative DNA polymerases, while the helicase continues to unwind the DNA leading to the generation of ssDNA which can be rapidly bound by the ssDNA binding protein RPA (replication protein A). In addition to replisome uncoupling, another, non-exclusive process is ssDNA produced by extensive DNA resection at arrested forks by specific exonucleases, such as Mre11 and Samhd1 (reviewed in [10]. Other obstacles that arrest progression of the entire replisome do not generate immediate ssDNA, and instead stimulate a process known as fork reversal. In this situation, replication forks backtrack to promote the annealing of the nascent DNA strands and protect it from degradation by nucleases. In order to resume DNA synthesis, the obstacles must either be removed or bypassed. Hence, cells can mobilize either of these mechanisms to reduce the frequency of stalled forks and preserve genome integrity. It is possible that a combination of these processes can occur upon replication forks stalling, depending upon the genomic context where the lesions are produced, and the nature of the DNA encountered by the replisome. While removal of the lesion is fulfilled by DNA repair, bypass occurs by activation of the DNA damage tolerance pathway (DDT), which includes TLS, template switch (TS), and repriming.

TLS is a pathway that cell can choose to reduce the frequency of stalled replication forks by allowing DNA damage tolerance and thus limiting the risk of fork collapse that can generate gross chromosomal rearrangement and threaten genome stability [11]. Stalling of replication forks is often caused by a variety of DNA lesions, as well as by secondary DNA structures that are difficult to replicate. Most DNA lesions that induce distortion of the DNA double helix stall the replicative DNA polymerases because their catalytic site cannot include the damaged base. In this situation, the ability to replicate altered DNA is ensured by TLS Pols, thanks to a large active site that can accommodate the distorted DNA bases. Although some TLS Pols are accurate at bypassing specific types of damage, such as Polη for UV-lesions, this process often occurs at the expense of DNA synthesis fidelity because these enzymes, which lack proofreading activity, exhibit low base discrimination, especially on undamaged DNA [12]. Due to its dangerous error-prone nature, TLS Pol recruitment is kept under strict control. This is highlighted by increasing findings of multiple regulatory mechanisms governing their stability and their association with the damaged and undamaged genome (see below). For example, Polη has been reported to be regulated at the post-translational level by phosphorylation, mono- and poly-ubiquitination, mono- and poly-sumoylation, as well as by OGTylation and by the tumor suppressor protein p53 at the transcriptional level (reviewed in [13]).

### 2.2. Recruitment and Action of TLS DNA Polymerases

Recruitment of Y-family TLS Pols (η, κ ι, and Rev1) at stalled forks depends upon the covalent linkage of an ubiquitin monomer on residue K164 on the replication fork component PCNA to generate monoubiquitinated PCNA (PCNA^mUb^, Figure 1) catalyzed by the (E3) ubiquitin ligase RAD18 in conjunction with its partner, the (E2) ligase RAD6 [14,15]. In theory, TLS Pol recruitment can occur on both lagging and leading strands of the replication fork. Notwithstanding, using an in vitro reconstituted system with recombinant yeast proteins, Polδ recruitment onto the leading strand has been shown to inhibit TLS [16], although it is not clear whether a similar situation may occur in vivo, under conditions of checkpoint activation. Altogether, these observations may suggest that TLS Pols could be mainly recruited to the lagging strand. This opens the question of how DNA lesions generated on the leading strand are managed. Recent observations have suggested that DNA lesion bypasses on the leading strand can be fulfilled by a newly discovered TLS Pol called Primpol. This enzyme has the unique ability to bypass DNA lesions as well as to reprime DNA synthesis downstream of the lesion [17]. Its recruitment depends upon the ssDNA binding protein RPA [18], and can therefore readily occur on the long stretches of ssDNA generated on the leading strand upon replication fork uncoupling. Repriming on the lagging strand can also occur downstream of the DNA lesion independently of TLS by the DNA Polα–primase complex which continuously reprimes on this strand for the generation of the Okazaki fragments. This mechanism is used as a default mechanism in yeast that does not have Primpol [19,20], while in metazoans, both bypass and repriming are likely to occur [21]. In both cases, repriming generates gaps behind the replication fork that will be filled-in at the end of S-phase and G2 by a gap-filling repair process. The use of these two pathways is not clear so far, but current models propose that bypass can operate on the lagging strand and repriming on the leading strand [22,23], although it is not clear when Primpol would reprime instead of bypassing. Two recent reports have shed some light on the interaction of these pathways by showing that repriming by Primpol appears to function as backup mechanism to restore replication when fork reversal is compromised [24,25].

TLS involves two critical events: bypass and DNA extension past the lesion [26]. Following nucleotide insertion across the lesion by a Y-family TLS Pol, the nascent strand, which is often mismatched, is extended by either Polκ or by the B-family polymerase, Pol ζ (made of Rev3–Rev7 subunits). Therefore, TLS requires the sequential action of two TLS Pols: the first inserts a nucleotide opposite the lesion, and the other one extends from it. It is currently unclear whether upon replication fork stalling, TLS Pols replace replicative Pols within the replisome, whether TLS Pols can coexist with replicative polymerases, or whether they are recruited at gaps left behind the replication fork. A number of observations suggest that these scenarios are all possible. Previous work in yeast has shown that the replisomes stalled by nucleotide shortage (hydroxyurea) are rather stable and do not dissociate from the DNA template [27]. However, the high affinity of replicative polymerase for primer–template (P/T) junctions may be altered when replisomes stall. The stalling process may induce conformational changes that increase the chances for TLS Pols to gain access to the stalled template through interaction with the processivity clamp, as recently suggested in vitro by an elegant work in *Escherichia coli* [28]. This work is, however, in contrast with a previous report using a single-molecule assay, showing that TLS Pol competition with replicative polymerase is instead concentration-dependent and is not affected by the lesion recognition [29]. This observation suggests that local protein concentration also contributes to TLS recruitment. In this respect, in mammalian cells, TLS Pol recruitment at DNA lesions generates nuclear foci, which are thought to be sites concentrating TLS factors, thereby facilitating lesion bypass [26]. In addition, activation of the ATR-dependent checkpoint may also contribute to conformational and/or structural changes in the replicative polymerases. For example, the p12 kDa subunit of the Polδ holoenzyme is degraded in an ATR-dependent manner through the concerted action of RNF8 and CRL4^Cdt2^ ubiquitin ligases [30]. Furthermore, Polζ interacts with the p50 and p66 Polδ subunits to form the Polζ4 complex [31,32,33]. It has been proposed that upon replication fork stalling, a shuffling of Polδ subunits with Polζ4 occurs. This complex facilitates Polζ tethering to the stalled replicative polymerase, and therefore positions it to facilitate the extension step of TLS, probably by displacing the Polδ catalytic subunit from the P/T junction (Figure 1). These observations suggest that the replication checkpoint may induce replisome remodeling to facilitate TLS at arrested forks. Co-existence is also possible. For example, in the early *Xenopus* embryo, at least one TLS Pol coexists within the replisome with the three replicative polymerases, making the replisome constantly DNA damage-tolerant [34]. In this particular context, the ATR-dependent checkpoint is largely inactive, supporting the possibility that by targeting replicative polymerases, this checkpoint may slow down DNA synthesis, which is unrestrained in the early embryo. 

Co-existence of different polymerases on PCNA is possible in principle because its homotrimeric structure can accommodate more than one protein on its binding surface. Indeed, PCNA is known to interact with many proteins, certainly not with all of them at one time, showing that PCNA is a flexible platform that can host a variety of factors involved in DNA replication, repair, and recombination [35]. This suggests that the interaction with all these factors must be orchestrated. In addition to PCNA post-translational modifications such as methylation and phosphorylation, recruitment of the CRL4^Cdt2^ (E3) ubiquitin ligase is another mechanism regulating PCNA partner switches. This enzyme catalyzes the polyubiquitination of a specific class of PCNA interactors containing a specialized PIP box, called PIP degron, including the CDK inhibitor p21 and the pre-replication factor Cdt1 [36]. Following DNA damage, CRL4^Cdt2^-dependent polyubiquitination and degradation of PIP degron-containing proteins allows PCNA to reset from a replication configuration to a DNA repair mode, including competence to carry out TLS [37] and Figure 1). This regulation provides one barrier to TLS pol access to the replisome when not needed.

### 2.3. Rad18 Chromatin Recruitment

Since the initial observation that Rad6/Rad18 promotes PCNA^mUb^ [40], several other factors have also been reported to contribute to this reaction, although their relationship with recruitment of the Rad6/Rad18 complex at stalled replication forks has not always been clarified.

Amongst the factors directly or indirectly implicated in recruitment of this complex to arrested forks, the first is ssDNA. Indeed, it was previously shown that replication fork uncoupling is important to induce PCNA^mUb^ [41]. Although Rad18 was originally shown to have affinity for ssDNA [42], previous [43] and more recent studies [44,45] propose that RPA also contributes to Rad18 recruitment. More investigations are required to clearly establish whether a physical complex between RPA and Rad6/Rad18 in vivo can be observed and whether RPA binding to ssDNA at stalled forks can recruit this complex. The Claspin component of the fork protection complex was also reported to be important for PCNA^mUb^ [46], probably by coordinating the replicative helicase with PCNA and mediating the interaction between PCNA and Rad18. The pre-mRNA splicing factor SART3 was also shown to be important to mediate Rad18–Polη interactions, as well as to generate ssDNA following DNA damage [47]. A very recent report in the yeast *Saccharomyces cerevisiae* proposes that the Rad52 protein, involved in homologous recombination, functions in stimulating PCNA^mUb^ independently of its HR function, probably by regulating the recruitment of RPA onto ssDNA gaps [48].

Interaction of Rad18 with TLS Polη and Polκ is also important for Rad18 recruitment to stalled forks [49]. In fact, Rad18 does not have a PCNA interaction domain (PIP box), while this is the case for TLS Polη and Polκ. Hence, Rad18 recruitment to stalled replication forks depends upon its interaction with TLS Pols. Consistent with this conclusion, Polη stabilization by overexpression of the USP7 ubiquitin hydrolase [50], or TLS Pols overexpression [51,52], leads to increased levels of PCNA^mUb^. On the same line, the PCNA-interacting protein SIVA1 was also shown to be important to bridge Rad18 interaction with PCNA and stimulate PCNA^mUb^ [53].

The DSB repair component of the MRX complex Nbs1 was also shown to be important to recruit Rad18 to DNA lesions through interaction with the Rad18 C-terminal domain that also binds Rad6 [52]. Furthermore, Rad18 was reported to interact with the ASK/DBF4/DRF1 regulatory subunit of the Cdc7 protein kinase [34,54]. Cdc7 has been involved in regulation of the DNA damage checkpoint [55], which becomes activated following replication fork uncoupling; therefore, it was proposed that Cdc7-ASK recruitment to stalled forks brings about Rad18, thus allowing lesion bypass [54]. ATR activation stabilizes Cdc7-ASK through inhibition of the APC ubiquitin ligase, thereby stabilizing a complex with Rad18. This observation provides an example of functional interaction between DNA damage response and DNA damage tolerance pathways. Finally, the cancer/ testes antigen MAGE-A4 was reported to interact and stabilize Rad18 to activate TLS [56].

The metalloprotease Spartan/DVC1 was previously described as a TLS regulator following UV damage in mammalian cells [57,58,59,60]. A functional homolog in yeast is the metalloprotease Wss1. Spartan contains a PCNA binding motif (PIP box) as well as a ubiquitin binding motif (UBZ), similar to that found in Rad18. Knock down of Spartan sensitizes cells to DNA damage and, interestingly, decreases binding of Rad18 to chromatin and, consequently, the level of PCNA^mUb^. Two other reports [60,61] also describe the identification of another Spartan binding protein, the p97/VCP/Cdc48 ubiquitin-selective chaperone implicated in various cellular processes, including proteasomal degradation and remodeling of protein complexes. It was shown that interaction of p97 with Spartan is important for the survival to DNA damage. Hence, Spartan, together with p97, could be important for the stabilization of PCNA^mUb^ and the final steps of TLS. In addition, [62] showed that Spartan also interacts with the replicative polymerase Polδ, suggesting a potential role in polymerase switch. Altogether, these data highlight an important requirement for Spartan in regulating Rad18 chromatin association and PCNA^mUb^ during the DNA damage response. USP1 (ubiquitin-specific peptidase 1) in complex with UAF1 (USP1-associated factor 1) is also important to regulate PCNA^mUb^. In response to UV irradiation, USP1 is degraded through auto cleavage, leading to the accumulation of PCNA^mUb^ [63]. 

Finally, very recently, the RFWD3 (E3) ubiquitin ligase has also been implicated in recruiting Rad18 (and probably other TLS factors) to UV-treated chromatin [64]. The authors propose that RFWD3-dependent ubiquitination of ssDNA-bound proteins may be a general mechanism to concentrate TLS factors and promote lesion bypass at stalled forks.

Collectively, these observations emphasizing the importance of Rad18 in the TLS process, highlight that several pathways contribute to Rad18 recruitment to stalled forks, including protein phosphorylation, ubiquitination, and chromatin remodeling (Figure 2). Whether these are independent modes of binding or cooperative functions remains to be elucidated. Another possibility is that these diverse factors mediate Rad6/Rad18 recruitment at stalled forks in relation to the nature of DNA damage and/or to the chromatin context where the replicative stress occurs. 

#### 2.3.1. Other TLS Pols Regulators

In addition to PCNA^mUb^, protein–protein interactions (PPIs), as well as TLS pol post-translational modifications (PTMs), are key modulators of TLS in mammalian cells. These modifications take place in space and time to limit mutagenic TLS activity on undamaged DNA and ensure optimal TLS at DNA lesions in rescuing stalled replication forks and thus preserve genome integrity [13]. Among these, Rad18 and PIAS1-dependent monoSUMOylation has been reported to regulate Polη association with replication forks encountering difficult-to-replicate DNA structures, thus preventing the occurrence of unreplicated DNA [65]. This observation is in line with previous reports showing that Polη is important to ensure DNA synthesis through difficult-to-replicate DNA regions [66,67]. Polη multi-site SUMOylation, at a different lysine residue by PIAS1, instead occurs upon DNA damage, followed by the action of SUMO-targeted ubiquitin ligases (STUbLs) to evict it from chromatin [68]. Taken together, these two observations suggest the existence of a SUMO-driven positive and negative feedback loop operating to regulate Polη association with replication forks and prevent unscheduled mutagenesis.

#### 2.3.2. Rad18 and DNA Repair

Apart from its role in TLS, Rad18 has been extensively shown to be implicated in homology directed DNA repair at DSBs [69,70]. This activity resides in the zinc finger and UBZ binding domain, which is distinct from the ring finger domain required for TLS [71,72]. Hence, Rad18 mutants that are TLS-deficient are fully HDR-proficient. The logic of this Rad18 function has so far been unclear, and it has been considered as an additional function uncoupled from TLS. A very recent observation from one of the authors (D.M.) has provided insight on another possible function of this activity. It has been observed that during the very early stage of *Xenopus* embryonic development, the HDR activity of Rad18 is coupled to its TLS activity, through PCNA^mUb^, thereby reducing the mutational load introduced. This mechanism adds up to the MMR activity that corrects mismatches introduced by TLS [73]. Taking into consideration this observation, it is possible that, in vivo, the mutagenesis rate of TLS Pols may be underestimated due to the concerted action of Rad18 repair activity. Whether this regulation also occurs in somatic cells is currently unknown.

TLS Pols are also implicated in the DNA replication-dependent repair of interstrand cross-links (ICLs). These lesions can be of endogenous nature generated by reactive aldehydes, and can also be generated upon exposure to chemotherapeutic agents, such as cisplatin and mitomycin C. In the embryonic *Xenopus cell-free*, in vitro system, restart of replication forks arrested at these DNA lesions requires replication fork convergence, in a process that involves Polζ-dependent bypass and extension as an error-prone process [74]. In mammalian cells, however, the CMG helicase has been reported to traverse these lesions, probably by remodeling of the CMG complex during transit through the lesion and assistance by other DNA helicases [9]. The nature of these differences is currently unclear. Analyses of proteins bound to ICL-treated chromatin in in vitro *Xenopus* egg extracts have also led to the identification of a number of factors required for replication-dependent repair. Amongst these were Rad18 in a complex with the SLF1 and SLF2 complexes, bridging the interaction with SMC5 and SMC6 subunits of the cohesion complex [75]. The Rad18–SLF1–SLF2 complex was shown to be important for homologous recombination-dependent DNA repair of ICL [75]. Furthermore, recent studies have shown that Spartan, by mediating DPC proteolysis, facilitates TLS past the lesion. In the absence of Spartan, DPCs are still degraded by the proteasome, but TLS across the resulting peptide adduct generated by the proteasome was likely not as efficient [76]. DPCs on the leading or lagging strand can pose impediments for helicase and/or polymerase progression; therefore, proteolysis removes the bulk of this obstacle and reduces the cross-linked protein to a peptide (remnant) that can be bypassed by TLS polymerase, thus resuming DNA replication [77]. These results may explain the significance of Spartan and Rad18 interaction and clarify this additional role for the TLS pathway in DPC repair.

### 2.4. TLS Inhibition in Cancer Therapy

RS is currently considered as a hallmark of cancer [78]. In some ways, cancer cells must become addicted to RS to allow proliferation. TLS Pols and TMEJ actively function in limiting excessive RS in cancer cells and allow their adaptation and progression. Previous work has shown that Rad18 and Polκ are important to confer tolerance to oncogene-induced replication stress [79]. Very recent observations show that oncogene-induced replication stress induces ssDNA gaps and slows down DNA synthesis. In this situation, TLS activation reverses these effects and facilitates the proliferation of transformed cells [80]. Curiously, protozoans such as *Leishmania* and *Trypanosoma*, which display high genomic plasticity and adaptation to toxicity induced by exposure to drugs, are characterized by the duplication of genes coding for TLS Pols [81]. It will be interesting to determine whether in these organisms this feature is functionally linked to genome dynamics and drug resistance. 

TLS is also involved in cancer resistance to therapeutic treatments, and several reports have shown that TLS Pols are overexpressed in cancer [3,82]. This can be achieved by the tolerance of DNA lesions introduced by chemotherapy and radiotherapy, with consequent increases in mutagenesis that can facilitate the selection of resistant tumor cells. Consistent with this possibility, a previous report showed that breast cancer cells cultured in 3D conditions, that more closely mimic a tumor microenvironment, upregulate TLS Pols in an ATR-dependent manner, compared to 2D culture, and showed increased cisplatin resistance [83]. Furthermore, Rad18 was shown to mediate resistance to ionizing radiation [84] and cisplatin [34] in glioblastoma cells. Moreover, a very recent report shows that Rad18 contributes to the mutational burden in a mouse model of induced skin tumors [85]. Furthermore, pharmacological inhibition of a Polζ–Rev1 complex improves melanoma chemotherapy [86], and a very recent report has interestingly shown that TLS Pols are upregulated in colorectal cancer resistant to BRAF and EGFR inhibitors, showing that TLS is a pathway that cells can activate to resist targeted therapy [87]. Along the same line, Polκ was very recently shown to be upregulated in melanoma, lung, and breast cancer cells treated with an inhibitor of the BRAF oncogene, contributing to drug resistance, probably by a non-mutagenic fashion [88], which may probably be through regulation of the DDR. In line with this possibility, a previous report has highlighted an important role of Polκ in sustaining resistance to the alkylating agent temozolomide in glioblastoma through activation of the ATR/Chk1 pathway [89], confirming a previous observation implicating Polκ in Chk1 phosphorylation [38]. Finally, upregulation of TLS Pols coupled with immunocheckpoint blockade has been reported to improve tumor regression upon treatment with cisplatin [90]. Altogether, these observations put TLS as an emerging appealing pathway to sensitize cancer cells to the therapy.

## 3. The Major Role of Polθ during the Alternative TMEJ Pathway to Repair DSBs

### 3.1. Several Pathways to Repair DSB

DSBs can arise under RS when DNA replication fork progression is altered by natural or pathological cellular events, or upon exposure to exogenous genotoxic agents such as ionizing radiation. DSBs compromise both genome integrity and cellular viability; therefore, their efficient and accurate repair is vital for cancer prevention and normal cellular proliferation. DSB repair pathways are broadly classified into two main types. The first one is homologous recombination (HR), which requires a 5′ to 3′ end resection, Rad51 loading, strand invasion, and DNA synthesis using an intact homologous template [91]. The second type includes the two end-joining (EJ) repair processes, the classical non homologous end joining NHEJ (c-NHEJ), and the alternative end joining (alt-EJ) pathways that do not require a homologous template. c-NHEJ is dependent on the Ku complex, DNA-PK, and XRCC4/Ligase 4 [92]. The alt-EJ pathway acts in a different way because it operates on a common resected HR intermediate. It starts with the search of short tracts of contiguous microhomology, followed by annealing steps, and then by a processing mechanism of 3′ ssDNA tails and DNA synthesis (Figure 2). The alt-EJ pathway includes the factors involved in the 5′ to 3′ resection (e.g., Mre11, Rad50, Nbs1, CtiP, and Exo1), PARP1, LIG3, as well as Polθ, a predominant mediator of this pathway [4,93]. The nucleolytic processing of broken DNA ends seems to be a central step in orientating the repair processes. Indeed, DNA end resection, which promotes repair by homologous recombination, can be inhibited by the chromatin-binding protein 53BP1, which shields DNA ends from nucleases through its ultimate effector Shieldin, a recently discovered four-subunit protein complex with ssDNA-binding activity, consisting of REV7 and the three proteins SHLD1, SHLD2, and SHLD3 which protects DNA ends and favors NHEJ [94]. This Shieldin complex is recruited to DSBs via SHLD3 in a 53BP1- and RIF1-dependent manner and constitutes a barrier to nucleases, thereby inhibiting resection. The role of TRIP13 ATPase, a negative regulator of REV7, in the dissociation of the REV7–Shieldin complex to promote HR has recently emerged as a critical regulator of DNA repair pathway choice [95].

### 3.2. The Critical Action of Polθ in TMEJ

The role of Polθ in TMEJ has been characterized in several multicellular organisms, such as worms, where it may promote genome diversification; fish and plants, where it is essential during the random integration of foreign DNA; as well as in mammals, where Polθ depletion sensitizes cells to several agents inducing DSB such as irradiation, bleomycin, etoposide, and camptothecin.

Polθ has an N-terminal helicase-like domain and a C-terminal A-family DNA polymerase domain [96] which seems to favor end joining of two separated DSBs (distal End Joining [6]). Structural analysis has revealed that Polθ can form dimers which may facilitate the proximity of DNA ends and stabilize synapsed intermediates [97] (Figure 3). Its activity is low in normal cells, and there is only a minor impact on development when POLQ is absent. In contrast, Polθ becomes critical in cells that are deficient of NHEJ or HR, including BRCA1/2 mutated cancer cells [7]. This observation is indicative of synthetically lethal genetic interactions between the backup Polθ/TMEJ repair pathway and NHEJ or HR pathways. Recent work has revealed a broader landscape of synthetic lethality with Polθ, emphasizing a critical and general role for Polθ in protecting cells from the accumulation of non-productive HR intermediates at sites of DNA replication-associated DSBs, even when canonical DSB repair pathways are functional [98]. In fact, unlike other DSBs, DSBs at collapsed forks are single ended, with no second end available for classical HR repair. These breaks can be processed by a replication-coupled repair named break-induced replication (BIR), a conservative DNA synthesis mechanism described as an HR-based repair pathway for one-ended DNA DSBs, which has been studied extensively in budding yeast [99] and more recently in mammalian cells [100]. BIR initiates at the broken forks, proceeds through the invasion by the broken chromosome of its homologous duplex, and moves as a D-loop along the length of the chromosome. At telomeres, similar processes of breaks in ALT cells seem to involve specialized DNA polymerases such as Polη [101] and Polλ [102]. In this particular context, TMEJ has been also proposed to stimulate the repair of broken forks [103]. TMEJ also functions when replication fork stalling and/or collapse are induced by G-quadruplexes [104], which are known to promote DSBs that cannot be repaired by HR. Indeed, worms lacking FANCJ, which is known to perform replication across G4 sites, rely on Polθ to repair collapsed forks and prevent the accumulation of large deletions. In addition, worms deficient of both Polη and Polκ accumulate DSBs at stalled forks that are repaired via a TMEJ mechanism [105]. This situation provides an example of diverse TLS pol crosstalk at stalled forks. Moreover, in human cells treated with the replication inhibitor hydroxyurea, Polθ depletion slows down replication fork progression and affects fork restart [103]). Direct evidence for a Polθ role at collapsed forks was recently provided using frog egg extracts [106]. The Polθ-helicase domain belongs to the SF2 family of helicases that comprises Hel308 and RecQ. Inhibition of the Polθ-helicase activity affects TMEJ in flies and chromosomal translocation in mouse cells. The Polθ-helicase domain has strong ATPase activity that is stimulated by ssDNA and DNA annealing activity, enabling base-pairing of ssDNA and facilitating TMEJ. Polθ also contains an exonuclease-like domain, although it lacks a 3′→5′ proofreading activity, explaining why Polθ is an error-prone polymerase [107]. Moreover, because of its low fidelity and the unique thumb domain that holds positively charged residues to grasp the unstable primer terminus, Polθ has the ability to extend DNA from mismatched primers [108]. Interestingly, it has been very recently discovered that Polθ can eliminate 3′ ends at microhomology regions prior to DNA synthesis [109], suggesting that Polθ holds an exonuclease activity that remodels DNA ends to facilitate their repair (Figure 3). In the future, it will be interesting to determine whether other exonucleases are also redundantly involved with Polθ in this process. 

In addition to its role during TMEJ, Polθ has additional repair functions in the repair of oxidative base damage [110] and mutagenic replication through UV lesions which prevents replication fork collapse, chromosomal instability, and UV-induced tumorigenesis [111].

### 3.3. Polθ Inhibition in Cancer Therapy

Expression of Polθ is tightly repressed in somatic cells, while it is upregulated in many human cancers, including breast, lung, gastric, and colon cancer. Polθ abundance is particularly high in HR-deficient breast and ovarian cancers and is associated with poor clinical outcomes. One of the underlying mechanisms that regulate Polθ expression has recently emerged in breast cancer cells. This process was reported to be linked to the zinc finger E-box binding homeobox 1 (ZEB1), a transcription factor inducer of the epithelial-to-mesenchymal transition (EMT), known to modulate breast cancer cell plasticity by conferring stemness properties to the cells [112]. ZEB1 has been demonstrated to interact directly with the POLQ promoter, to control the expression of the POLQ gene and prevent TMEJ activity [113].

In the TMEJ process, a bidirectional scanning process by Polθ initiated from the 3′ termini can identify microhomologies, allow annealing and subsequent DNA polymerization [114]. Due to its poor processivity, aborted synthesis happens frequently, triggering additional rounds of microhomology search, annealing, and DNA synthesis, resulting in insertions of DNA sequences that are identical to flanking DNA. These templated insertions (TINS), whose frequency correlates with POLQ expression levels, can reach up to seven TINS/genome in BRCA mutated cancer genomes, higher than breast cancer genomes with no germline BRCA mutations (two TINS/genome) [114]. Despite these mutagenic features, Polθ can preserve genome stability by preventing chromosome translocations [6]. It is then possible that high abundance of Polθ observed in multiple cancers, which often correlates with poor patient survival [115,116,117], could help cancer cells to limit excessive RS and genome instability and prevent large, catastrophic deletions. The mutational synthesis associated to such an adaptive mechanism could further reinforce diversity and evolutionary advantage. 

The reliance of BRCA mutated cells on TMEJ for survival renders this unique polymerase an attractive target for cancer treatment. In fact, a great deal of studies are currently in progress to develop Polθ inhibitors for the treatment of HR defective tumors. Exploiting synthetic lethality in DNA repair to eliminate cancer cells is best illustrated by the PARP1–BRCA genetic interaction. The significant efficiency of PARP1 inhibition on the survival of HR defective tumors has led to the development of several PARP inhibitors that are currently in clinical trials, and clinical benefits have been achieved in BRCA-mutated ovarian, breast, and prostate cancers. Unfortunately, resistance to PARP inhibitors is frequently observed in the clinic, and besides the well-documented reversion of BRCA mutations that can restore HR activity, the mechanisms underlying such resistance are actively being investigated. Polθ might emerge from these studies as a strong possible candidate.

## 4. Conclusions

Cells have evolved mechanisms to minimize chromosome rearrangements when DNA synthesis is impeded by naturally occurring obstacles or by exogenous cues, such as exposure to environmental chemicals or cancer therapeutic treatments. Rad18-mediated TLS and TMEJ constitute two important safeguard pathways that cells can employ to avoid gross chromosome rearrangements at stalled replication forks. Rad18 and Y-family TLS Pols facilitate DNA lesion bypass so as to avoid DSBs generated following replication fork collapse, as a result of persistent stalling of DNA synthesis. TMEJ is a vital back-up pathway that allows DNA repair at broken replication forks, which minimizes gross chromosomal rearrangement when both HR-dependent pathways and TLS are inefficient. Both pathways are error-prone; therefore, their occurrence inevitably generates mutations. This is a toll to pay for survival, but one that cancer cell can exploit to adapt to severe natural selection conditions, such as those imposed by heavy therapeutic treatments. Thus, targeting either Rad18-mediated TLS or Polθ has the potential to substantially improve clinical outcomes in cancer. Inhibiting Y-family TLS Pols and/or Rad18 may contribute to reducing both the endogenous and chemotherapy-induced mutational burdens of tumors. Rad18 inhibition may, at the same time, reduce cancer fitness by decreasing RS tolerance and increasing endogenous DNA damage such as ssDNA gaps. These structures have recently been proposed to cause the BRCA cancer phenotype, “BRCAness”, and determine the mechanism of action of genotoxic chemotherapies as well as PARPi synthetic lethality [80,118,119]. Polθ inhibition may be important to eliminate a critical back-up DNA repair mechanism on which HR-deficient tumors rely to survive to chemotherapy. Future research in these issues may lead to the development of more efficient therapeutic treatments to improve patient life expectancy.

## Figures and Tables

**Figure 1 ijms-22-03924-f001:**
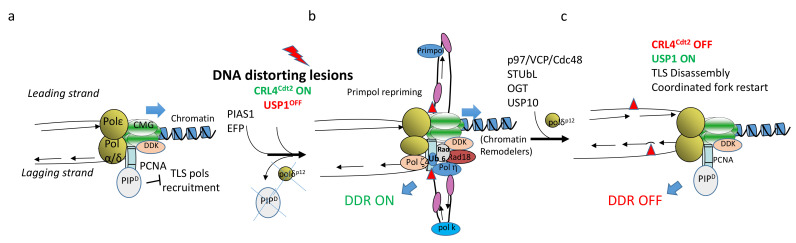
Speculative model of translesion synthesis (TLS) pol recruitment at replication forks stalled by DNA distorting lesions. (**a**) When the replisome encounters DNA distorting lesions (red triangles), replicative polymerases (Pol α, δ, ε) are stalled, while the replicative helicase (CMG complex) can slide through. This process generates an excess of ssDNA (**b**). Discontinuous DNA synthesis continues on the lagging strand, which generates small replication intermediates that prime the recruitment of specific checkpoint factors leading to activation of the ATR-dependent checkpoint (DDR, not shown). TLS Polκ is recruited onto these replication intermediates to stabilize them and facilitate ATR signaling through Chk1 phosphorylation [38]. DDR activation activates the CRL4^Cdt2^ ubiquitin ligase, which induces PCNA-dependent PIP degron-containing proteins destruction, such as the p12 subunit of the Polδ holoenzyme. This process facilitates Polζ4 complex recruitment into the Polδ holoenzyme to position it for bypass extension. Recruitment of RPA onto ssDNA brings about Primpol and other factors (see Figure 2). Primpol allows repriming on the leading strand. Rad18 binding is also stabilized by interaction with Polη and DDK. The Rad6/Rad18 complex then catalyzes PCNA^mUb^, thus stabilizing interaction of Polη (and/or other Y-family TLS pols) with PCNA. Polη poly-SUMO modification by PIAS1 and PCNA ISGYlation by the ISG15 (E3) ligase EPF also occurs at this stage. Following this step, the lesion is b-bypassed by the sequential action of Polη and Polζ (insertion and extension, respectively). Upon lesion bypass, Polη is polyubiquitinated by CRL4^Cdt2^, OGTylated, and extracted from chromatin by the p97/VPC/Segregase complex to be targeted for proteasomal degradation, while PCNA is deubiquitinated by the concerted action of USP1 and USP10, thus leading to disassembly of the TLS complex including Rad18 (**c**). The Ubr5 ubiquitin ligase is also implicated in this step by mediating Polη chromatin binding through the ubiquitination of histone H2B [39]. Finally, inactivation of CRL4^Cdt2^ and stabilization of the ubiquitin hydrolase USP1 resets PCNA, to a DNA replication-competent form followed by TLS Pol degradation and/or release, thus allowing fork restart.

**Figure 2 ijms-22-03924-f002:**
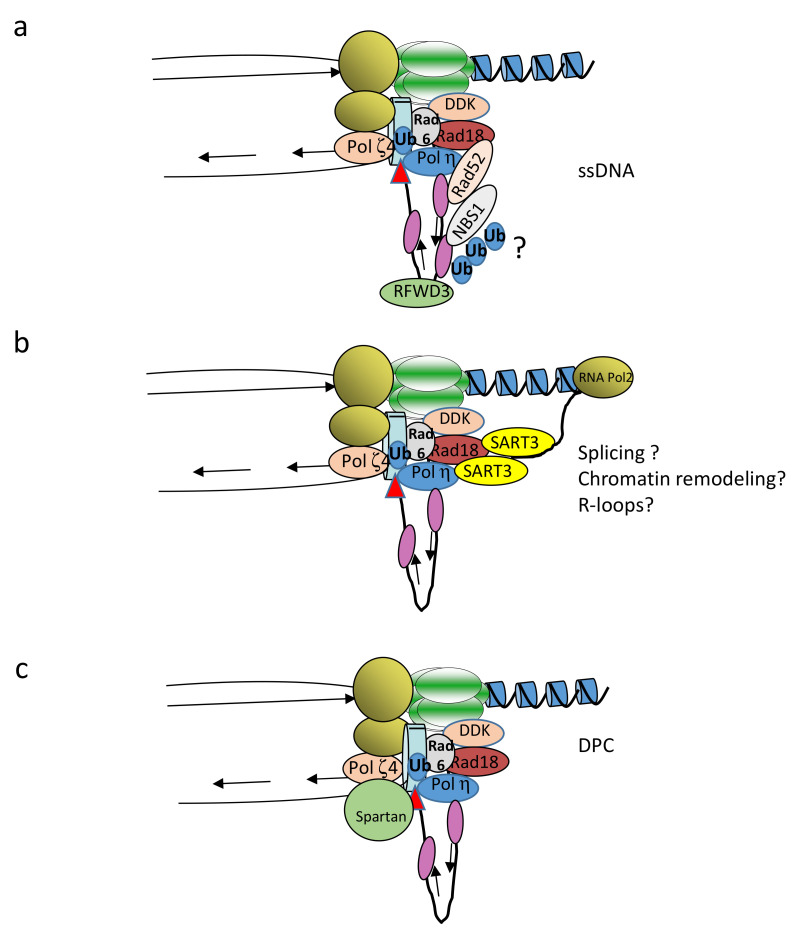
Factors mediating Rad18 recruitment. (**a**) Recruitment of Rad52, NBS1 and the RFWD3 ubiquitin ligase occurs onto ssDNA via replication protein A (RPA). Rad52 and NBS1 may bridge the interaction between RPA and Rad18, while RFWD3 might non-specifically polyubiquitinate proteins bound to ssDNA (question mark), thus facilitating TLS factor nucleation. (**b**) The SART3 pre-mRNA splicing factor may facilitate chromatin remodeling and PCNA^mUb^ through interaction with Rad18 and Polη when replication forks encounter transcription units (RNA pol2). RNA–DNA hybrids with displaced ssDNA (R-loops) may also be generated in this situation, and thus promote RPA and Rad18 recruitment. (**c**) DNA–protein crosslinks (DPC) stimulate recruitment of the Spartan metalloprotease through interaction with PCNA. Spartan recruitment will facilitate Rad18 recruitment and PCNA^mUb^.

**Figure 3 ijms-22-03924-f003:**
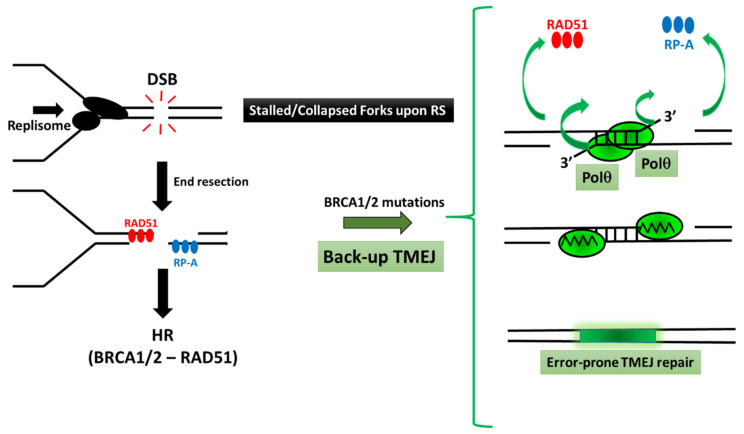
Speculative model of theta-mediated end joining (TMEJ)/Polθ activity as back-up repair of DSB in homologous recombination (HR) deficient cells. DSBs occurring at stalled/collapsed replication forks in S/G2 phases of the cell cycle can be resected leading to short segments of ssDNA which are quickly coated by RPA and can be exchanged for RAD51 when long end-resection occurs, then promoting strand invasion and copying from the sister chromatid in a proficient BRCA-dependent HR pathway. When HR is defective (BRCA mutated genes), or when alternative repair is preferred (choice poorly understood), TMEJ can act as a back-up repair pathway, involving the critical role of Polθ. Polθ-helicase can displace either RPA or RAD51, and Polθ polymerase promotes the synapsis of the opposing ends and performs a bidirectional scanning initiated from the 3′ termini to identify internal microhomologies which can be annealed, thus generating 3′ flaps. Polθ and/or FEN1 can remove the 3′ flaps, and Polθ can start the repair DNA synthesis with poor processivity and frequent aborted synthesis, resulting in a high rate of mutations.

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
