# Peer review of "Translesion Synthesis or Repair by Specialized DNA Polymerases Limits Excessive Genomic Instability upon Replication Stress"

_ijms, 2021, doi:10.3390/ijms22083924_

Round 1
Reviewer 1 Report
Maiorano et al. “TLS or Repair by specialized DNA polymerases to limit excessive genetic instability upon replicative stress”
In this manuscript the authors try to review the contribution of specialized DNA polymerases, such as TLS pols, in limiting genetic instability when DNA replication faces unrepaired DNA damage in cells. In this scenario, these enzymes are decisive to either avoid broken forks or favor their repair after collapse. An attractive idea pointed out by the authors is that although the genetic instability that characterizes cancer cells is necessary as a driver of cell viability and development of resistance to anticancer treatments, an excess of such instability can cause cell death and hence it must be limited. This is where the relevance of DNA damage bypass mediated by TLS pols or a specialized repair such as pol theta-mediated end joining comes into play. It is undoubtedly an attractive and topical subject, of great interest both in basic biology and biomedicine, and the review would be very useful.
As it is written, it is not easy to find the connection between the two main topics (TLS pols and pol theta-mediated repair) in the manuscript. The first block is well structured and, despite being mainly focused on pol eta, it is very consistent. The Rad18 part is quite comprehensive, although perhaps some irrelevant details could be removed and the clarity of others enhanced. In my opinion, in some places there is an excessive enumeration of features that can lead a non-expert reader (for whom this type of reviews should be directed) to a lack of understanding. I have tried to identify some of these parts in my comments listed below. Should the authors agree with my suggestions, I think the manuscript will gain in clarity. Likewise, regarding this first topic, I think that the manuscript would gain also in clarity if the figure 1 were split into two or more sub-figures. The current figure 1 is overloaded with content, and is difficult to follow; in my opinion the information it is intended to reflect there is too complex to be adequately covered and explained in a single figure. In the second part (the one related to pol theta and its role in the repair of DSBs), I do not see clearly the relationship with what the review is about according to its title (replicative stress). Perhaps the authors should alleviate some text referring to the general repair mechanisms of DSBs, which can “dilute” a bit that relationship that has been hard for me to see. Finally, I have missed some comment or even hypothesis about the possible contribution of any of these specialized polymerases in one of the aspects less discussed in the text (but not less relevant) such as the formation and/or development of reversed forks. In addition to these general comments, I have been able to find some errors or statements that could lead to misinterpretation by non-expert readers, which I detail below.
Comments:
Page 1, line 38-40
“Accurate replication of undamaged DNA involves the action of the “replicative” “high-fidelity” DNA polymerases Pol alpha, epsilon and delta which have proofreading ability (Hoffmann and Cazaux, 2010)”.
Why do the authors use quotation marks for such widespread and accepted concepts as that these polymerases are replicative or that they have a high synthesis fidelity? The quotes should be removed (in fact, they are removed later in the text). On the other hand, it is a mistake to consider that pol alpha has proofreading activity because it does not have it. Also, the reference indicated in this paragraph does not seem the most appropriate, since it refers to non-replicative polymerases (“alternative, specialized or TLS (translesion synthesis) DNA polymerases”)
Page 2, line 81-83
“Replication forks frequently encounter obstacles that slow down or even halt the progression of DNA synthesis leading to RS. The obstacles can be of endogenous nature (oxidized bases, ribonucleotides), in addition to …”
I find it difficult to think these types of lesions (oxidized bases, ribonucleotides) as the best examples of DNA lesions that can cause a slow down or even arrest of DNA synthesis progression. In fact, it has just been published that the eukaryotic replisome fully tolerates oxidative damage present in DNA => https://www.embopress.org/doi/full/10.15252/embj.2020107037
Page 3, line 143-44
“Although PCNA can be recruited on the leading strand and promote TLS Pols loading upon monoubiquitination, …”
As has been written by the authors here, it appears that PCNA is not permanently associated with pol epsilon during leading strand synthesis. As far as I know, the most widespread replisome models imply that PCNA travels in the replisome on both lagging and leading strands. If there are alternative models, it should be noted here.
Page 4, line 149
The statement “… although it is not clear when Primpol would reprime instead of bypassing and repriming” is a bit confusing for me in the context in which is introduced. I would understand that Primpol either does one thing (bypass) or does the other (reprime), right? Perhaps the authors want to indicate that “… although it is not clear when Primpol would reprime instead of bypassing” ? Or is it a misunderstanding on my part?
Page 4, line 165
Why do the authors exclude Primpol here, if they just indicated that it is also a TLS?
Page 5, line 217
In the section dedicated to "Rad18 chromatin recruitment" there are several things that, in principle, do not have much to do with the epigraph and, in my opinion, mess up the understanding of the topic. For example, it is the case of introducing Claspin (coordination between helicase/PCNA), the HR-independent function of Rad52, or the recruitment of Rad18 by Nbs1 … I would restrict this section to the detailed description of factors directly involved in Rad18 recruitment to stalled forks in the context of replicative stress, avoiding converting this in a messy enumeration of direct or indirect factors that makes it difficult to clearly understand the final message. On the other hand, in my opinion, the essential fact here is that Rad18 does not have a domain of direct interaction with PCNA, hence it is necessary to describe what are the possible ways that exist to be recruited to blocked RFs; there is the key to introducing this section in the manuscript.
Similarly, despite all the reported information about the protease Spartan in this section, it is not clear to me in the end what its role is in promoting Rad18 recruitment to chromatin during RS response, and the consequent mono-ubiquitination of PCNA. In fact, Spartan recognizes PCNA when it is already ubiquitinated, which would lead to think that its role happens downstream with respect to the recruitment of Rad18. This should be clarified, even before introducing other factors (such as p97) that more than clarify, generate more confusion.
Given the large number of factors that can influence Rad18 recruitment to chromatin in a scenario of RS, perhaps a specific figure summarizing it would be a good idea, which also would reduce the excessive complexity of current figure 1.
Page 7, line 313
“In addition to PCNAmUb, protein-protein interactions (PPIs), as well as post-translational modifications (PTMs), are key modulators of TLS in mammalian cells”.
Indeed, mono-ubiquitination is a PTM
Page 7, lines 328-342
I really do not understand what is the contribution of the paragraph "Rad18 and DNA repair" in this review on the role of specialized DNA polymerases in TLS and repair.
Page 7, line 350
Perhaps the authors consider it appropriate to comment here on the novel role of Primpol in the replication traverse process, despite this still unpublished in a peer-reviewed journal
https://www.biorxiv.org/content/10.1101/2020.05.19.104729v1
Page 8, lines 371-373
The authors should note that not all TLS events are error-prone.TLS carried out by certain TLS pols over their cognate lesions can be relatively accurate, as happens with Pol eta over a cyclobutane thymine-thymine dimers. On the other hand, there is a repetitive sentence here:
“TLS is also involved in cancer resistance to therapeutic treatments … (371)”
“TLS is also important to overcome cancer therapeutic treatment leading to resistance … (373)”
Page 8-9
The description of the DSBs repair mechanisms has already been introduced previously in the manuscript. In any case, as I have commented before, in my opinion this information is irrelevant in this review and the reader can be referred to any of the many good texts that have been previously published anywhere.
Page 9, lines 444-445
The authors should not miss the opportunity to include at this point the possibility that single-ended DSBs can be repaired via break induced replication. Despite BIR has been largely studied in yeast, there are recent studies (e.g. https://pubmed.ncbi.nlm.nih.gov/33470420/) that implicate this pathway in the repair of seDSBs derived from replicative stress in mammalian cells too. It would be very appropriate in this review to discuss the possible competition between both mechanisms, BIR and TMEJ.
Page 10, lines 471-473
Some reference to this observation should be included
Page 11, lines 495
The reference to work by Prodhomme et al. can be updated
Page 11, lines 496-510
In my opinion, the text corresponding to these lines does not fit into the “Polθ inhibition in cancer therapy” section, but rather the previous one, which describes how pol theta works in TMEJ. Likewise, in that section there are issues (such as the SL of pol theta and the BRCA genes) that perhaps would fit more in the final section. The authors could rearrange the text a bit in these sections.
Other minor points:
- I think the acronym TLS in the title sounds weird and it should be changed for its meaning "Translesion Synthesis"
- There are some typographical errors throughout the text (e.g. line 31 “sub clones”, line 528 “by-pass”)
- Excessive (and inappropriate) use of quotation marks (“replicative”, “high-fidelity”, “flexibility”). It distracts. In fact, "replicative" or "high-fidelity" is used in quotes at the beginning and then it becomes normal, without quotes (eg in line 170).
- I wonder why is so much of the work carried out by the Prakash lab, one of the pioneers in the TLS pols fields, omitted in this manuscript? In my opinion, there should be more reference to their work. In this sense, several papers are known where the role of pol theta has been demonstrated in the context of TLS beyond the role in TMEJ and, even a recent publication also demonstrates the role of another specialized polymerase, namely Pol lambda, in this complex scenario. Perhaps the authors should mention some of these studies in their manuscript.

Reviewer 2 Report
In this review manuscript, Maiorano et al. present a summary of the current state of knowledge regarding the involvement of Rad18 and Polq in the replication stress response (RSR) and how the cancer cells able to activate those two pathways can benefit from a survival advantage by mitigating the adverse events of excessive replication stress (RS). The topic of this manuscript is very “hot” since there has been a lot of important advances in this specific branch of the DNA repair field lately; most notably, the recent discovery by the Cantor lab challenging the current assumption that double-strand breaks (DSBs) are the reason for chemosensitivity in tumors with “BRCAness” and proposing that it is rather the replication associated single-stranded DNA (ssDNA) gaps that cause chemosensitivity, with TLS playing a critical role in replication gaps suppression (GS) for high cancer cell fitness (Nayak et al 2020, Panzarino et al., 2021 and Cong et al., BioRxiv 2019).
This review manuscript from Maiorano et al. represents an exhaustive work and is globally well written. This review will be of high interest to people working on the replication stress response. Despite the manuscript being very thorough, below are a few weaknesses that, if fixed, could improve the quality of the manuscript before publication.
Minor concerns
- I would suggest adjusting the title to “TLS or Repair by specialized DNA polymerases limits excessive genomic instability upon replication stress” or something like “Specialized DNA polymerases (can) limit excessive replication stress-induced genomic instability by TLS or repair synthesis”
- Please add a sentence to introduce Rad18 or Polq in the abstract, since those are the main two topics of the review.
- Please clarify the sentence lines 54-55, about chromosome duplication: do you mean reversed stalled forks or something else?
- Please clarify the sentence lines 73-74 as it is too wordy or just delete and combine with the next sentence which seems to say the same.
- The description of the 3 different consequences of RS is quite confusing the way it is explained. I do not think those 3 scenari (fork uncoupling, extensive DNA resection and fork reversal) depend on the nature of the encountered obstacle, as it is stated lines 91/92. Fork uncoupling activates the ATR-mediated checkpoint response, fork reversal allow for fork protection and DNA repair, while excessive (not extensive) resection is a pathological situation that might happen when the fork is not properly protected (i.e Fork Protection [FP] function of BRCA1/2). If talking of normal short range resection for HR, then it would be good to talk about DSB repair through HR.
- Around lines 106/7, it would be good to briefly discuss the ATR-mediated local versus global fork reversal mechanism after interstrand cross-links (ICLs) induction described by the work of Lopes and colleagues (Mutreja et al., 2018).
- In the second paragraph of section 2.1, it should also be acknowledged that some TLS Pol are accurate at bypassing specific types of damage, such as Polh for UV-lesions during replication (Sale et al., 2012, cited in the paper {83}).
- Could you please try to improve the flow and focus on critical details in paragraph 2.2 as follows?
- Mention that PCNA is mono-ubiquitinated on residue K164 (line 132)
- PCNA is not an “accessory” factor (line 134) but an essential one!
- As it is generally accepted in the field and well summarized by Sale et al., 2012 “fully effective activity by all DNA polymerases requires them to be tethered to the DNA through interaction with the sliding clamp PCNA”, so I do not fully understand the point about the difference of TLS between the leading and lagging strand. It comes very abruptly and breaks the flow, which is already hard to follow in that section.
- Line 152: would be worth of mentioning the term “Okazaki fragments”
- Line 160: two recent reports have...
- Please discuss the findings of Dungrawala et al. 2015 (your ref {26}) about replisome stability versus dissociation upon HU treatment around lines 173-174.
- Please correct the misunderstanding that PIP degrons are proteins: PIP degron is not a protein, it is in itself a degradation motif associated with PIP motif, the motif for PCNA binding (line 211).
- Paragraph 2.3:
- The results lines 232-233 are not commented: what does it mean that a splicing factor is involved?
- Rad18 does not have a PIP box (line 240), how about Rad6?
- The paragraph about Rad18 recruitment through Pol eta and kappa is a bit provocative, especially when placing it before any mention of UBM or UBZ motifs that are present in all Y-family polymerases, besides their PIP boxes. Aren’t the TLS Pol supposed to be recruited only when PCNA is monoubiquitinated? Please clarify this discordance.
- Please include a discussion on the balance between USP1 and Rad18 activities for controlling PCNA monoUb levels, as well as the post-translational and p97 AAA ATPase-mediated controls of the RSR. Those points have been recently reviewed in Rageul et al., DNA Repair 2019.
- “A very recent observation”... if this is unpublished data, please mention it in brackets.
- Line 343: I guess you mean interstrand cross-links and not intrastrand.
- Line 346: mitomycin C is the correct spelling.
- Similarly to TLS being involved in ICL repair, TLS is also involved in DNA-Protein Crosslink (DPC) repair. Please discuss this using the papers from Walter’s and Duxin’s labs (Sparks et al., Cell 2019 and Larsen et al., 2019).
- However, I wonder whether there is actual evidence for a role of RAD18 itself in those specific TLS contexts (ICL or DPC)? Please clarify.
- Section 3 is very well written and flowing better than section 2. I just have 3 comments:
- The function and regulation of REV7 in DSBs repair choice have been illuminated by the work of D’Andrea’s lab in Clairmont et al., Nature Cell Biology 2020. Please include a brief comment/discussion about it at the end of section 3.1.
- The new findings of the Cantor and Cimprich labs (Panzarino et al., 2021 and Cong et al., BioRxiv 2019) should be touched upon along with the “Nayak et al 2020” citation line 539/540.
- Figure 2 represents a double-ended DSB, shouldn’t it be a one-ended DSB (after the passage of the fork)?
- Typos/English mistakes/formatting:
- Line 15: change “needed to adapt to selection pressures” for “necessary to adapt to selective pressures”?
- Line 33: enhanced => improved
- Line 34: extreme => extremely
- Lots of “s” missing at verbs conjugated at 3rd person singular... (lines 66, 183, 218, 365, 463)
- Line 73 and possibly other instances: genetic instability => genomic instability
- Line 87: as well as its nature (remove “by”)
- Line 101: backtrack to promote...
- Line 125: post-translational level
- Line 238: remove “was”
- Description of PRR acronym in the Legend of Figure 1? + remove description of the red triangles lines 291/292 because it is a repeat of line 290.
- Line 318: has => have
- Line 350: during transit through the lesion
- Line 392: resistance
- Line 414: Nbs1?
- Line 430: such as...
- Lines 437 and 455: a period is missing at the end of the sentence
- Lines 453 and 284: diverse
- Line 533: for survival, but one that cancer cell...
